# Piezoelectric Sensor-Embedded Smart Rock for Damage Monitoring in a Prestressed Anchorage Zone

**DOI:** 10.3390/s21020353

**Published:** 2021-01-07

**Authors:** Quang-Quang Pham, Ngoc-Loi Dang, Jeong-Tae Kim

**Affiliations:** Department of Ocean Engineering, Pukyong National University, Busan 48723, Korea; bkdn06x3a@gmail.com (Q.-Q.P.); loi.ngocdang@gmail.com (N.-L.D.)

**Keywords:** smart rock, electromechanical impedance, impedance-based technique, concrete damage, anchorage zone

## Abstract

In this paper, a piezoelectric sensor-embedded smart rock is proposed for the electromechanical impedance monitoring of internal concrete damage in a prestressed anchorage zone. Firstly, a piezoelectric sensor-embedded smart rock is analyzed for impedance monitoring in concrete structures. An impedance measurement model is analyzed for the PZT (lead zirconate titanate)-embedded smart rock under compression in a concrete member. Secondly, a prototype of the smart rock embedded with a PZT sensor is designed in order to ascertain, sensitively, the variations of the impedance signatures induced by concrete damage in an anchorage zone. Thirdly, the performance of the smart rock is estimated from a numerical analysis and experimental tests. Variations in the impedance signals under compressive test cases are analyzed in order to predetermine the sensitive frequency band for the impedance monitoring. Lastly, an experiment on an anchorage zone embedded with the smart rocks and surface-mounted PZT sensors is conducted for the impedance measurement under a series of loading cases. The impedance variations are quantified in order to comparatively evaluate the feasibility of the sensor-embedded smart rock for the detection of internal concrete damage in the anchorage zone. The results show that the internal concrete damage was successfully detected using the PZT-embedded smart rock, thus enabling the application of the technique for anchorage zone health monitoring.

## 1. Introduction

Prestressed concrete (PSC) has been widely used as critical members for bridge construction, thanks to its cost-effectiveness and crack-resistance over original reinforced concrete structures. In PSC bridges, anchorage zones have a vital function in carrying the designed prestress-force and transferring it into the structures. Considering time-dependent effects, the structural performance of PSC members could gradually decline due to material deterioration, corrosion, and time-dependent prestress loss, leading to long-term deformation. Incipient concrete damage, which is often in the form of inner micro-cracks, could produce and amalgamate to induce surface cracks. The cracks often occur in tensile regions, and most cracks are invisible at an incipient state. When the inner cracks propagate onto the PSC members’ surfaces, material degradations could be significant due to environmental erosion. The severe concrete cracks in PSC bridges cause a rising concern for designers, investigators, and managers [1,2,3]. Thus, the detection of concrete cracks in PSC structures at the early stage is crucial in order to ensure structural integrity and reduce long-term maintenance costs.

Various SHM (structural health monitoring) methods have been proposed for the damage detection of PSC members using vibration characteristics (i.e., natural frequencies, mode shapes, and modal curvatures) [4,5]. The methods utilize low modal parameters, which are conspicuously insensitive to incipient damage. Moreover, several local techniques, such as visual inspections and X-rays, have been implemented for the monitoring of the structural damage of PSC structures [6,7,8]. Although the methods are recognized in several applications, their sensitivities are inadequate for the development of a reliable detection of the incipient damage (e.g., internal concrete damage) or material deteriorations, in addition to their being time-consuming, costly, and even unsafe for the inspectors. Strain-based methods have also been used to detect structural damage by utilizing the well-established relationship between stress and strain. Among various strain sensors, fiber optic sensors have been commonly used for damage monitoring [9,10] due to their high durability and sensitivity. Nonetheless, the fabrication of sensing-cable–embedded fiber optic sensors remains a challenge. Furthermore, the hybrid health monitoring algorithms using the combination of the global dynamic characteristics (e.g., changes in natural frequencies) and local responses (e.g., changes in acoustic emission features) yielded an effective SHM system [11,12,13]. Lacidogna et al. [11] proposed multi-technique damage detection for a plain notched-concrete beam using modal frequencies for alarming damage occurrence and the combined acoustic emissions and digital image correlation for the estimation of the damage evaluation. The results show the effectiveness of the evaluation of the surface crack development in notched beams.

In the past decades, various SHM applications have been successfully implemented using the functions of smart materials such as piezoelectric ceramic. A PZT (e.g., lead zirconate titanate) patch is attached to a monitored structure in order to acquire electromechanical impedance responses, and then the changes in the impedance signals are quantified and used as a damage indication [14,15,16]. Ai et al. [17] and Narayanan and Subramaniam [18] attached PZT sensors to concrete members in order to monitor concrete crack-induced overloading. However, since the impedance features of surface-mounted sensors are quite sensitive to environmental changes [19], this method requires a complex technique to compensate for temperature effects [20,21]. In order to overcome the issue, the PZT-embedded concrete block (a so-called ‘smart aggregate’ [22]) has been employed to monitor concrete structures. Several applications have been carried out in order to evaluate the applicability of the smart aggregate to the SHM of concrete members [22,23,24]. However, the predetermination of the frequency bands for the impedance monitoring of the smart aggregate has not been conducted so far.

In PSC bridges, anchorage zones resist the particularly high compressive force induced by prestressing strands [25] during the steel strands’ installation and life-cycle structures. Under various operation conditions (e.g., increments of traffic volume and material degradation) and environmental effects (e.g., thermal expansions and natural disaster events), the anchorage zones could be potentially damaged. Incipient damage often occurs in the form of micro-cracks before propagating onto the surfaces of the anchorages, because anchorage systems are usually embedded in concrete structures [26,27]. In order to ensure the structural integrity of the anchorage zones, some design methods and new anchorage types have been proposed [28,29,30]. Moreover, many studies have been conducted on the assessment of the health conditions of prestressing strands using impedance features obtained from PZT sensors [31,32]. However, the implementation of the impedance-based technique for the detection of the internal concrete damage of anchorage zones has not been significantly conducted. The major advantages of the impedance-based method are as follow: (1) PZT sensors are embeddable, and thus directly receive stress variation in an inspected structure because more stress variations yield more changes in the impedance features [14,33]; (2) it is a cost-effective method due to the use of various low-cost forms and the light weight of PZT sensors, enabling the damage localization using an array of PZT sensors [31]; and (3) it allows online and real-time monitoring capacity [34,35].

This study presents an impedance-based damage detection method in a concrete anchorage zone via a PZT-embedded smart rock. Firstly, an impedance measurement model is designed for the impedance monitoring of a concrete member using the PZT-embedded smart rock. Secondly, a prototype design of the smart rock embedded with a PZT sensor is proposed in order to sensitively catch the impedance variations induced by concrete damage in an anchorage zone. Thirdly, the performance of the smart rock is tested using numerical analysis and experimental tests. The variations in the impedance signals under compressive loading cases are analyzed in order to predetermine the sensitive frequency band for the impedance monitoring. Lastly, an experiment on an anchorage zone embedded with the smart rocks and surface-mounted PZT sensors is conducted for the impedance measurement under a series of loading cases. The PZT’s impedance variations are quantified in order to comparatively evaluate the feasibility of smart rock for the detection of internal concrete damage in the anchorage zone.

## 2. Theoretical Model of the Piezoelectric Sensor-Embedded Smart Rock

### 2.1. Impedance Measurement Model of the PZT-Embedded Smart Rock

Figure 1 shows an impedance measurement model proposed for concrete members using a PZT-embedded smart rock. The smart rock is formed by embedding a protected PZT patch (e.g., epoxy) into a small concrete block before casting. Then, the smart rock is positioned in a monitored structure in order to acquire the impedance responses of the coupling interactions between the PZT smart rock and the host structure (see Figure 1a). When the force *F* on the host structure changes, it leads to changes in the modal parameters (e.g., stiffness and damping ratio) of the protected layer, the concrete block, and the host structure. As shown in Figure 1b, the 3-degrees of freedom (3-DOFs) impedance model represents the coupled motions of the protective layer, the concrete block, and the host structure. The parameters *m*, *k*, and *c* symbolize the mass, the damping coefficient, and the spring stiffness, respectively, and the subscripts *p*, *r*, and *s* denote the protective layer, the concrete block, and the concrete structure, respectively.

For the 3-DOF impedance model, the governing equation of the motion under a harmonic excitation *F_p_* at the PZT driving point can be formulated as the following:(1)$$\left[\begin{array}{ccc} m_{p} & 0 & 0\\ 0 & m_{r} & 0\\ 0 & 0 & m_{s} \end{array}\right]\left\{\begin{array}{c} {\ddot{x}}_{p}\\ {\ddot{x}}_{r}\\ {\ddot{x}}_{s} \end{array}\right\}+\left[\begin{array}{ccc} c_{p} & -c_{p} & 0\\ -c_{p} & c_{p}+c_{r} & -c_{r}\\ 0 & -c_{r} & c_{r}+c_{s} \end{array}\right]\left\{\begin{array}{c} {\dot{x}}_{p}\\ {\dot{x}}_{r}\\ {\dot{x}}_{s} \end{array}\right\}+\left[\begin{array}{ccc} k_{p} & -k_{p} & 0\\ -k_{p} & k_{p}+k_{r} & -k_{r}\\ 0 & -k_{r} & k_{r}+k_{s} \end{array}\right]\left\{\begin{array}{c} x_{p}\\ x_{r}\\ x_{s} \end{array}\right\}=\left\{\begin{array}{c} F_{p}\\ 0\\ 0 \end{array}\right\}$$
where ${\ddot{x}}_{p},{\dot{x}}_{p},x_{p}$; ${\ddot{x}}_{r},{\dot{x}}_{r},x_{r}$; and ${\ddot{x}}_{s},{\dot{x}}_{s},x_{s}$ are the accelerations, the velocities, and the displacements corresponding to the motions of masses *m_p_*, *m_r_*, and *m_s_*, respectively. The coupling structural–mechanical (SM) impedance, the *Z_s_* of the protected layer, the concrete block, and the host structure are determined as follows [36]:(2)$$Z_{s}=\frac{1}{j\omega }\left(K_{11}-\frac{K_{12}^{2}}{K_{11}+\left(K_{22}-K_{23}^{2}/K_{33}\right)}\right)$$
where the terms *K_mn_* (*m,n* = 1–3) are the dynamic stiffness coefficients that contain the structural parameters of the protective layer, the concrete block, and the monitored structure [36].

The electromechanical impedance, *Z*(*ω*), is a function of the SM impedance, *Z_s_,* of the smart rock host structure, and that of the PZT sensor, *Z_a_* [37]:(3)$$Z(\omega )={\left\{j\omega \frac{w_{a}l_{a}}{t_{a}}\left[{\hat{\varepsilon }}_{33}^{T}-\frac{1}{Z_{a}(\omega )/Z_{s}(\omega )+1}d_{31}^{2}{\hat{Y}}_{11}^{E}\right]\right\}}^{-1}$$
where *w_a_*, *l_a_* and *t_a_* are the width, length, and thickness of the piezoelectric patch, respectively; ${\hat{\varepsilon }}_{33}^{T}$ is the complex dielectric constant at zero stress; $Z_{a}\left(\omega \right)={\hat{Y}}_{11}^{E}w_{a}t_{a}/\left(j\omega l_{a}\right)$ is the SM impedance of the PZT patch; *d*_31_ is the piezoelectric constant in one direction at zero stress; and ${\hat{Y}}_{11}^{E}$ denotes the complex Young’s modulus of the PZT patch at the zero electric field.

As noted in Equation (3), the real-part impedance responses *Z*(*ω*) contain the SM impedance of the PZT patch, the smart rock, and the host structure. The 3-DOF impedance model represents the coupled vibrations of the smart rock and the host structure. When the electric and mechanical properties of the PZT sensor are constant, any external effects (e.g., concrete damage, force change) would exert an effect on its impedance responses.

### 2.2. Mechanism of the PZT-Embedded Smart Rock under Compression

Figure 2 shows the stress responses and corresponding impedance signatures of a PZT-embedded smart rock during the strength development of concrete. The coated PZT has no stress acting on the cover layer of the PZT, and it is under compression *σ_o_*_1_ due to the strength development of the concrete (see Figure 2a). The smart rock is embedded in a concrete structure before casting, such that the smart rock is under compression *σ_o_*_2_ while the coated PZT is under *σ_o_*_1_
*+ σ_o_*_2_ (see Figure 2b). Previous studies have shown that the hydration process of a cubic concrete sample embedded with PZT sensors [38,39] caused rightward shifts in frequencies and decreased in impedance magnitudes. Thus, the impedance responses of the PZT-embedded smart rock before the application of external loads are illustrated in Figure 2c.

Figure 3 shows the stress responses and corresponding impedance signals of the PZT-embedded smart rock in the concrete structure under compressive force *F*. The smart rock’s surfaces are under compression *σ_F_* along with the longitudinal direction (i.e., *z*-axis) and under tension *νσ_F_* (due to Poisson’s effect, in which *ν* is Poisson’s ratio) for the other surfaces, as seen in Figure 3a. As a result, the compressive stresses on the smart rock (*σ_o_*_2_
*+ σ_F_*) and the coated PZT (*σ_o_*_1_
*+ σ_o_*_2_
*+ σ_F_*) are increased for the *z*-direction, but they are reduced for the other directions. For concrete members under compression, inner concrete damage (often in the form of cracks) could occur before surface cracks [40,41]. The behavior of a concrete member can be classified into two states: an intact state (no damage occurrence) and a damage state (inner concrete damage, crack propagation, and failure). In the intact state, the impedance signals are changed rarely due to the prestress acting on the coated PZT (i.e., *σ_o_*_1_
*+ σ_o_*_2_) [24]. When inner concrete cracks occur locally close to the coated PZT sensor, the prestress is partially released. This reveals that the boundary conditions surrounding the coated PZT are changed, resulting in rushed changes in the impedance responses of the embedded PZT sensors [24,40]. Figure 3b illustrates the impedance responses of the PZT-embedded smart rock in a concrete member under compression.

### 2.3. Statistical Quantification of the Impedance Responses

In order to quantify the changes in the impedance signatures via the PZT-embedded smart rock, the root mean square deviation (RMSD) damage metric is commonly [42] used:(4)$$RMSD\left(Z,Z^{*}\right)=\sqrt{\left(\sum_{i=1}^{N}{\left[Z^{*}(\omega_{i})-Z(\omega_{i})\right]}^{2}\right)/\sum_{i=1}^{N}{\left[Z(\omega_{i})\right]}^{2}}$$
where *Z*(*ω_i_*) and *Z^*^*(*ω_i_*) are the real part of impedance signatures obtained from the intact and loading states of the smart rock at the *i*th frequency, respectively, and *N* denotes the number of frequency sampling points in the sweep. The RMSD metric, which represents the intensity of random spectra, is useful in order to quantify the variation of the impedance signals in a frequency band.

Ideally, the RMSD value should be larger than zero if the compression forces increase (i.e., the stress variation); otherwise, it should be zero. However, the statistical index of the RMSD can be larger than zero due to experimental and computational errors, even if there are no changes in the applied forces. The upper control limit (UCL) [43] is established for decision making on the stress variations as follows:(5)UCL=μ+3σ
where *μ* and *σ* are the mean and the standard deviation of the impedance signatures in the intact condition, respectively. The UCL was calculated by 3*σ*, which corresponds to a confidence level of 99.7%. The calculation of the UCL is outlined in three steps: firstly, *N* impedance signals are measured at the intact condition; secondly, the RMSD values for *N* impedance signals are computed by Equation (4), then a set of RMSD values is obtained; finally, the UCL of the RMSD values is calculated using Equation (5). The occurrence of damage (e.g., the change of stress) is indicated when the RMSD values are larger than the UCL. Otherwise, there is no indication of damage occurrence.

## 3. PZT-Embedded Smart Rock for Impedance-Based Concrete Damage Detection

### 3.1. Prototype Design of the PZT-Embedded Smart Rock

Since most piezoelectric materials (e.g., PZT patch) are fragile, they are commonly covered in a smart aggregate, which is a coated PZT embedded into a small concrete block [22,44] before its positioning in a concrete member in order to monitor structural performance during the concrete’s curing [45] and service time [16]. For the impedance-based SHM methods, the monitoring results mainly depend on the selections of effective frequency bands [46,47]. Meanwhile, the sensitive frequency bands are dependent on the material properties and geometric sizes of the smart aggregate’s components [24,48]. In general, the sensitive frequency band relies on the orientation of the sensors in the target structures. The predetermination of the sensitive frequency ranges could help to minimize the time for the impedance-based monitoring technique in practices, and to detect incipient damage in a monitored structure [13,31].

As shown in Figure 4, a PZT-embedded smart rock was designed for the structural health monitoring of concrete structures. The PZT 5A patch was 10 mm × 10 mm × 1 mm in size, and it was welded with an electric wire in order to ascertain the impedance responses. Furthermore, the PZT patch was covered by 0.5 mm thick epoxy in order to waterproof it and insulate it from electricity (see Figure 4a). Then, the protected PZT was placed in a small concrete block (φ 26 mm and H = 26 mm) before casting (see Figure 4b). The concrete mixture for the anchorage zone is listed in Table 1. As noted in the table, the size of the smart rock is simplified as D_max_ of the aggregate. A concrete mixture including cement, sand, and water (without aggregate) was used for the fabrication of the smart rocks. Figure 4c shows the samples of PZT-embedded smart rock at 24 h after the removal of the frameworks. The smart rocks were used to analyze the realistic performance under compression, as described in Section 3.2 and Section 3.3. They were embedded into a concrete anchorage zone (detailed in Section 4) for impedance-based damage monitoring.

Figure 5 shows the impedance spectra of naked PZT, coated PZT (covered with epoxy resin at 12 h and 20 h), and PZT embedded in concrete (smart rock). This figure shows the information on how the process affects the PZT’s impedance signals. Peak 1’s impedance of the naked PZT was 191.5 kHz, and it had the highest real impedance magnitudes. The impedance frequency and its magnitude were reduced when the PZT was coated with the epoxy. This observation was consistent with the previous study [40]. During the development of the strength of the epoxy and concrete, Peak 1’s impedance reduced in magnitude and increased in frequency. Conversely, the impedance signals of the smart rock would be increased when the concrete is damaged and the constrain condition of the PZT patch is released, indicating that the PZT-embedded smart rock would disclose the occurrence of damage in the concrete.

### 3.2. Numerical Analysis of Smart Rock-Based Impedance Monitoring

#### 3.2.1. Finite Element Model of PZT-Embedded Smart Rock

The sensitivity of the impedance responses obtained from a PZT-embedded smart rock could be affected by the orientation of the PZT sensors in an inspected structure. In order to deal with this issue, two finite element (FE) models (Models 1–2) of the smart rock under the effects of the axial and lateral compressive force were established in Comsol Multiphysics (Comsol Inc., Burlington, MA, USA), as shown in Figure 6a,b, respectively. For the effect of the axial force, a force *P* was applied on the smart rock’s surface along the axial direction (i.e., *z*-axis) (see Figure 6a). For the effect of the lateral force, the compressive force *P* was placed upon 2.6 mm × 26 mm (0.1 φ × H [49]) of the smart rock’s surface (see Figure 6b).

The FE models consist of the PZT 5A patch (namely PZT-A for the axial force effect and PZT-L for the lateral force effect), the epoxy layer, and the concrete block. The geometric parameters of each component were previously described in Section 3.1. The material properties of the PZT sensor are listed in Table 2 [14]. The material properties of the concrete block and the epoxy layer [50] are listed in Table 3. The compressive strength of concrete was 23.3 MPa for the concrete mixture, as given in Table 1 (tested on φ 15 cm and H = 30 cm). Most concrete has a compressive strength of more than 28 MPa; the compressive strength of the smart rock (see Figure 4c) was a little lower. Since the concrete mix of smart rock was fabricated using the same ratio of cement, sand, and water as the concrete mixture of the anchorage zone, its material properties were assumed to have the same properties of the main structure (see Table 3). For the impedance analysis, also, the material properties of the smart rock’s components were assumed to be linear elastic materials.

For Model 1, the complete meshed FE model of the smart rock consists of 10,972 elements, including 9964 elements for the concrete block, 752 elements for the epoxy layer, and 256 elements for the PZT. The fixed boundary condition was assigned to the bottom surface of the concrete block (see Figure 6a). For Model 2, the FE model consists of 10,634 elements. The fixed boundary condition was assigned on the opposite surface, as shown in Figure 6b. The quadratic tetrahedral elements were used for the FE models.

Seven cases (P1–P7) of the compressive forces were simulated in a series in order to obtain the impedance responses of the PZT-embedded smart rock, as follows: P1 (1.0 kN), P2 (1.5 kN), P3 (2.0 kN), P4 (2.5 kN), P5 (3.0 kN), P6 (3.5 kN), and P7 (4.0 kN), corresponding to axial stress change from 0–5.65 MPa (i.e., P1–P7). The forces P1–P7 could cause lateral stress changes from 0–3.8 MPa, in which the lateral stress could be computed as 2P/(πHD) [49] (with φ = 26 mm and H = 26 mm in Figure 6). The axial stress variation was slightly different from the lateral stress under the same applied forces. However, when the smart rock is embedded in the structure, the concrete of the smart rock’s host structure is unified. The force *P* on the host structure causes the same stress variations for a local region close to the PZT sensor, and it is not affected by the shape of the smart rock. In order to acquire the impedance signatures, a 1 V harmonic voltage was applied to the top surface of the PZT patch, while the bottom surface was assigned as the ground electrode.

#### 3.2.2. Numerical Impedance Responses of the PZT-Embedded Smart Rock

Figure 7a,b shows the impedance responses of PZT-A (under axial force) and PZT-L (under lateral force) in the frequency range of 100–600 kHz under the compressive forces P1–P7. There are three main resonant frequencies, which were about 200 kHz for Peak 1, 245 kHz for Peak 2, and 460 kHz for Peak 3. The resonant impedance frequency peaks were slightly different due to the dissimilarity in the boundary conditions of the smart rocks in the two FE models. Among the three peaks, Peak 1’s impedance shows the most variation, but the changes in the impedance peaks were insignificant.

It is known that the resonant impedance frequency ranges represent meaningful structural information [14,51]. The frequency range of 100–250 kHz (including Peak 1’s impedance) was selected in order to quantify the variations in the impedance signals induced by the compressive forces. The RMSD indices of PZT-A and PZT-L were computed for the selected frequency range under the forces P1–P7, as depicted in Figure 8. The RMSD indices were increased linearly under increasing compressive forces, and the PZT-A’s RMSD indices are slightly larger than those of PZT-L.

### 3.3. Experimental Analysis of Smart Rock-Based Impedance Monitoring

#### 3.3.1. Experimental Set-Up on the Compressive Tester

In order to analyze the realistic performance of the PZT-embedded smart rock (in Section 3.2), an experimental test of the smart rocks was carried out, as illustrated in Figure 9. A smart rock was placed between two plates of a compressive machine (see Figure 9a). A load cell was used to measure the real applied forces. The loading speed was set to 0.1 mm/min in order to introduce compressive forces to the smart rock. Four tests (Tests 1–4) on the three smart rocks—namely SR-1, SR-2, and SR-3—were conducted for the impedance measurement, as listed in Table 4. The smart rocks SR-1 and SR-2 were tested under axial compressive forces (Test 1) and lateral compressive forces (Test 2). For the smart rock SR-3, first, tests on the effects of axial forces (Test 3) on the smart rock’s impedance responses were conducted. After Test 3, SR-3 was removed and kept in free boundary conditions for about two hours. Then, the effects of the lateral forces (Test 4) were tested. The geometric parameters and material properties of the smart rocks were described in Section 3.1.

For each test (Tests 1–4), the compressive force was increased from 1.0 to 4.0 kN (P1–P7) with an interval of 0.5 kN, as noted in Table 4. In order to set the same boundary conditions for the smart rocks, the force was set to 1.0 kN (a baseline). An impedance analyzer (HIOKI 3532, HIOKI E.E. Corporation, Nagano, Japan) was used to excite (using 1 V harmonic excitation) and record the impedance responses in the range of 100–600 kHz (500 intervals) for the smart rocks. The temperature variation measured via KYOWA (EDX-100A, Kyowa Electronic Instrument Co., Ltd., Tokyo, Japan) was less than 1 °C during the impedance measurement. As such, temperature effects on the impedance responses could be ignored in this analysis. Furthermore, in order to minimize the effects of electromagnetic noises on the impedance signals induced by flexing, twisting, or transient impacts on a coaxial cable, the cable was placed in such a way that the cable was not affected by anything dynamic during the impedance measurements. Four ensembles of impedance signals were recorded for each case of the compressive force in order to compute the control threshold (UCL shown in Equation (5)) to make a decision on the damage occurrence and to estimate the effects of noise on the impedance responses.

#### 3.3.2. Experimental Impedance Responses of the PZT-Embedded Smart Rock under Compression

Figure 10a,b, shows the measured impedance spectra (in the frequency range of 100–600 kHz) of SR-1 and SR-2, respectively, under seven loading cases (i.e., P1–P7). The three resonant impedance signals (i.e., the three peaks in the figure) were measured as follows: about 198 kHz for Peak 1, 250 kHz for Peak 2, and 465 kHz for Peak 3. The impedance responses of SR-3 under the axial and lateral compressive forces were zoomed in at the frequency range of 150–250 kHz, as seen in Figure 10c. In the intact case (P1), Peak 1’s impedance of smart rock SR-3 was 182.5 kHz in Test 3 (under the effect of axial load) and 182.7 kHz in Test 4 (under the effect of lateral force). The impedance frequency had an ignorable change, thus indicating that the smart rock would not be damaged before Test 4. Peak 1’s impedance signatures show high real-impedance responses and the most variation under compressive force among the three impedance peaks. Thus, the frequency range of 150–250 kHz was chosen in order to quantify the impedance responses induced by the force variations.

Figure 11 shows the RMSD indices of the impedance signals of SR-1, SR-2, and SR-3 (computed for the frequency range 150–250 kHz) under loading cases P1–P7. The control threshold UCLs were established using the measured signals at force P1 (the baseline signal). The error bars calculated from four ensembles of the impedance signals were also depicted. As observed in the figure, the magnitudes of the RMSD were ignorable in the intact case, but they were beyond the UCLs for the loading cases P2–P7, suggesting that the variations of the compressive forces were successfully alarmed. The standard deviations of impedance signals were comparatively small (see error bars in the figures), suggesting a low dispersion of the data. In other words, the electromagnetic effects on the impedance features could be ignored. Moreover, the RMSD indices of SR-1 versus SR-3 (under axial force) and SR-2 versus SR-3 (under lateral force) were linearly increased and comparable to each other. These results demonstrated the feasibility of the PZT-embedded smart rock by consistently measuring the impedance features.

### 3.4. Sensitivity of the Impedance Responses via PZT-Embedded Smart Rock to Stress Variation

Figure 12 shows a comparison of the numerical and experimental impedance frequencies for Peaks 1–3 under axial and lateral compressive forces. The differences in the impedance frequencies are relatively small (less than 3% for SR-1 and SR-2). Meanwhile, Peak 1’s frequency of SR-3 was about 7% different from those of SR-1, SR-2, or the numerical result. Furthermore, the impedance spectra of SR-3 had a broader bandwidth than those of the other smart rocks. The differences could come from an inaccurate thickness of the epoxy layer among the coated PZT sensors. The previous study showed that the thickness of the adhesive layer had significant impedance responses [48,52]. Despite that, the FE tool was applicable for the analysis of the impedance responses of the PZT-embedded smart rock.

Figure 13 shows a comparison of the rates of the RMSD’s numerical (see Figure 8) and experimental (see Figure 11) impedance for the effects of axial and lateral forces. The RMSD index at force P7 was selected as the reference for the computation. The relationship between the applied forces and the RMSD indices were well-matched for the smart rocks under the axial forces (see Figure 13a) and the lateral forces (Figure 13b), except for some errors at SR-2 (under the lateral force). Furthermore, there were inconsistencies in the magnitudes of the RSMD indices between the numerical (see Figure 8) and experimental analyses (see Figure 13). The reasons could be: (1) the effect of the prestress *σ_o_*_1_ due to the hydration process of the concrete acting on the coated PZT (see Figure 2a) was not be considered in the FE analysis, and (2) the natures of the concrete and the epoxy layer could not be linear under compressive force.

From the analyses, the following observations were made: (1) there were slight differences in the impedance features under the loading-direction effects; (2) the impedance feature was repeatable, and was gradually increased under compressive forces; and (3) the predetermined frequency band for the smart rock was 100–600 kHz, in which the sensitive frequency band was 150–250 kHz.

## 4. Feasibility Evaluation of PZT-Embedded Smart Rock for Concrete Damage Detection in Prestressed Anchorage Zone

### 4.1. Test Set-Up and Test Scenarios

#### 4.1.1. Experimental Set-Up on the Anchorage Zone

A lab-scale experiment on a concrete anchorage zone equipped with smart rocks was conducted in order to evaluate the feasibility of the PZT-embedded smart rock for the health monitoring of the anchorage zone, as schematized in Figure 14. The anchorage zone includes a concrete block (25 cm × 25 cm × 32 cm) and four steel-blocks (3.8 cm × 3.8 cm × 3.8 cm), which played a role as a bearing plate [53] in order to transfer the compressive forces to the structure. The steel blocks were evenly distributed on the concrete surface, and their distance to the center of the anchorage was 6.5 cm (see Figure 14b). A cylinder (φ 110 mm, made of PVC, Polyvinyl chloride) played a role as a duct for the cables (see Figure 14a). For the feasibility evaluation of the smart rock, the anchorage zone was designed without reinforcement. The material properties of the concrete are listed in Table 3, with a compressive strength *σ_c_* = 23.3 MPa.

Two smart rocks, namely SR-A (on the left side) and SR-B (on the right side), were embedded in the concrete anchorage (3.5 cm from the concrete surface) for the impedance monitoring, as detailed in Figure 14a–c. The geometries and material properties of the smart rocks are presented in Section 3. Furthermore, two surface-mounted PZT sensors, namely PZT-S1 (close to SR-A) and PZT-S2 (close to SR-B), were utilized for the impedance measurement, in order to compare their sensitivities with the embedded sensors. The surface-mounted PZT sensors had a size of 10 mm × 10 mm × 0.5 mm, and they were attached via aluminum interfaces (10 mm × 10 mm × 0.5 mm) before being mounted onto the anchorage’s surface. The use of the aluminum interface could enhance the sensitivity of the impedance responses [54]. PZT5A also was selected for the surface-mounted PZTs sensors with material properties, as shown in Table 2.

Figure 14c shows the set-up of the anchorage zone on a supporting steel frame. The steel frame with a capacity of 300 kN was designed to resist the tension forces introduced by the anchorage zone. Two thick steel plates (steel plates 1–2) were connected by four steel-tubes using bolted connections. The compressive force was introduced to the anchorage zone using a hydraulic jack, and a load cell was used to record the real applied forces.

#### 4.1.2. Test Scenarios for Impedance Measurement in the Anchorage Zone

Table 5 shows 11 loading cases (F1–F11) simulated for the impedance measurement in the anchorage zone. At first, the compressive force was set to F1 = 100 kN, as a baseline. For the first six loading cases (i.e., F1–F6), the applied forces were gradually increased from 100 to 200 kN, with an increment of 20 kN for each case. For the next three loading cases (i.e., F6–F9), the applied forces were progressively raised from 200 to 230 kN, with an interval of 10 kN. Following that, the force was increased to F10 (235 kN). The concrete was partially collapsed under the force F10, and then the compressive force was reduced to F11 (102 kN).

As noted in Table 5, the bearing stress *σ_b_* was varied from 17.3–40.7 MPa (i.e., 0.74–1.74 *σ_c_*) under the forces F1–F11. It is well known that the bearing stress in the anchorage zone (with reinforcement) allows from 1.25 to 3.0 *σ_ci_* [41,55,56], in which *σ_ci_* (≈0.85 *σ_c_*) is the compressive strength of the concrete at steel strand’s installation. In the experiment, the concrete surface crack (near SR-A) occurred under the bearing stress *σ_b_* = 1.71 *σ_c_* (i.e., under force F9), and the anchorage failure occurred under *σ_b_* = 1.74 *σ*_c_ (under force F10). The observation of the concrete’s crack development during the loading procedure is summarized in Table 5, and is also illustrated in Figure 15. There was no crack when the force was lower than 220 kN (i.e., the force F8), as seen in Figure 15(a1,b1). When the force was increased to F9, an incipient crack was found only on the left surface of the anchorage zone (close to SR-A), as seen in Figure 15(a2,b2). When the force was increased to F10, cracks occurred and propagated (mostly surrounding SR-A), and then the concrete failed first in a region close to SR-A (see Figure 15(a3)). As a result, the larger cracks occurred (close to SR-B, see Figure 15(b3)), and the concrete failed after the other side was damaged. A Dazor 8MC-100 magnifier was used to check the crack occurrence during the loading cases.

An impedance analyzer HIOKI 3532 was used to generate (1 V harmonic voltage) and measure the impedance signatures of the PZT sensors in the frequency range of 100–600 kHz (501 points). For each test case, four ensembles of impedance signals were recorded. As noted in Table 5, the impedance signals of SR-A, SR-B, and PZT-S1 were measured during the crack propagation under force F10. Meanwhile, PZT-S2 was not measured due to the sudden collapse of the anchorage zone, which came from the lack of reinforcement. During the impedance measurement, the room temperature was controlled at about 22 °C by utilizing air conditioners to minimize the influence of temperature variations on the impedance features.

### 4.2. Damage Monitoring Results on the Anchorage Zone using the Smart Rock

#### 4.2.1. Impedance Responses of PZT-Embedded Smart Rocks and Surface-Mounted PZTs

Figure 16 shows the impedance responses of the smart rocks SR-A and SR-B (in the frequency range 100–600 kHz) under the compressive forces F1–F11. The three resonant impedance signals (i.e., the three peaks in the figure) were measured as follows: about 210 kHz for Peak 1, 265 kHz for Peak 2, and 470 kHz for Peak 3. Among these, Peak 1’s impedance signals are more sensitive to force variations.

For SR-A (see Figure 16a), the impedance signals were insignificantly varied under forces F1–F6. The impedance signals were suddenly changed under force F7, thus revealing the transformation of the concrete medium around SR-A (e.g., inner concrete damage [40]). The impedance responses were continuously altered under force F8, and it underwent another abrupt change under force F9 (an incipient surface crack close to SR-A). The impedance signals varied under force F10 (crack propagation), and they showed the most variation under force F11 (concrete failure). For SR-B (see Figure 16b), the impedance signatures showed insignificant variations under forces F1–F10. The substantial alteration only occurred as the anchorage zone collapsed (under force F11).

Figure 17 shows the impedance responses of the surface-mounted PZT-S1 and PZT-S2 (in the frequency range of 100–600 kHz) under the eleven loading cases. There were several resonant peaks in the examined ranges, and Peak 1’s impedance also showed the most variation. For PZT-S1 (see Figure 17a), the impedance signals showed insignificant changes under forces F1–F8. The impedance signals suddenly changed under force F9. The signals kept changing under forces F10–F11. For PZT-S2 (see Figure 17b), the impedance signatures showed ignorable changes under forces F1–F10, and showed a rapid change when the anchorage zone was partially damaged (the force F11).

By matching the impedance signatures of the embedded-PZT sensors and the surface-mounted PZT sensors to the crack appearances, the observations were made as follows. First, the crack occurrence caused rushed changes in the impedance responses; that is, SR-A and PZT-S1’s impedance showed significant changes, but those of SR-B and PZT-S2 were ignorable changes (under the force F9). Second, the incipient crack and damage in the concrete, which early occurred on the left side of the anchorage zone, could come from the unequally-applied force (via four steel blocks) and the nature of the concrete structure (i.e., an inelastic material). In Figure 16a, a clear peak frequency shift is observed, and the bandwidth also shifts.

#### 4.2.2. Damage Monitoring in the Anchorage Zone Using PZT’s Impedance Features

As observed in Figure 16 and Figure 17, Peak 1’s impedance signals show the most variation, so the frequency range of 100–250 kHz was selected for the impedance-based damage detection. The RMSD index was used to quantify impedance changes, and the control threshold UCL was utilized for decision-making. Figure 18 shows the RSMD indices of SR-A and SR-B calculated for 11 loading cases. Furthermore, the error bars calculated from the four ensembles of the impedance signals are also depicted. The RSMD magnitudes were ignorable (0.2%) under the intact case, and they gradually increased with respect to the loading cases. The standard deviations of impedance signals were comparatively small (see error bar in the figures), suggesting the low dispersion of the data.

For SR-A (see Figure 18a), the RMSD indices were insignificant for forces F1–F6 (i.e., 0.2–4.3%). Under force F7, the RMSD index (i.e., 7.9%) became significant (about twice as large as that under force F6), suggesting sudden changes in the concrete surrounding SR-A. Under force F9, the RMSD index (29.5%) was about three times that under force F8. The change could be caused by the inner damage that instigated the incipient the surface crack. Then, the RMSD value was the maximum (i.e., 117.0%) under force F11. For SR-B (see Figure 18b), the RMSD values were insignificantly small (0.2–3.2%) under forces F1–F10. Under force F11, the RMSD index (i.e., 19.8%) became six times higher than those under force F10. The results again confirmed that the variations of the PZT-embedded smart rocks’ impedance signatures were induced by the concrete damage. 

Figure 19 shows the RSMD indices of PZT-S1 and PZT-S2 computed for the 11 loading cases. The RSMD magnitudes were ignorable (i.e., 0.2%) in the intact case. For PZT-S1 (see Figure 19a), the RMSD values (0.2–8.7%) gradually increased under forces F1–F10, and the value (i.e., 6.1%) was reduced under force F11. For PZT-S2 (see Figure 19b), the RMSD values were stably small (<2.0%) under forces F1–F9, and the RMSD index was a significant value (15.0%) when the concrete was damaged.

#### 4.2.3. Discussion of the Damage Monitoring on the Anchorage Zone using Smart Rock

When the concrete was damaged, the impedance responses progressively increased in their magnitudes and were reduced in their impedance frequencies (see Figure 16a), thus evidencing the mechanism of the PZT-embedded smart rock (as described in Section 2). The rapid changes in the impedance signatures revealed that the concrete domain was damaged. Thus, the anchorage zone was classified into the two states (see Figure 18a): an intact state (for force lower than F6), and a damaged state, starting with the incipient damage (under force F7), the damage propagation (under force F8), the initial surface crack (under force F9), and the concrete failure (under force F10).

The impedance features of the PZT-embedded smart rock yielded a better damage indicator than those of the surface-mounted PZT sensor. This is because the PZT-embedded smart rocks directly received the stress variations from the anchorage zone, as opposed to the surface-mounted sensors. Previous studies [33,57] have demonstrated that the more stress changes yielded, the more variations in the impedance responses. As a result, the embedded PZT sensors yielded higher sensitivities in their impedance responses. Furthermore, it is known that PZT’s impedance responses are insensitive to compressive forces [33,58], and the variations in the impedance features came from changes in the boundary conditions [33] or stiffness reduction [58]. In this study, the impedance responses of the surface-mounted sensors were also relatively less sensitive when the compressive force was lower than F6 (intact case, see Figure 19a). The impedance signals of PZT-S1 (close to SR-A) show relatively higher sensitivity than those of PZT-S2. This could be due to unequal applied forces (via the four steel blocks). When the forces increased (larger than F6) they could cause changes in the boundary conditions (e.g., concrete damage) surrounding the sensors, leading to the differences in their impedance features.

## 5. Concluding Remarks

In this paper, impedance-based damage detection in the anchorage zone using impedance features measured by PZT-embedded smart rock was presented. Firstly, the impedance measurement model was designed for the impedance monitoring of the concrete member using the PZT-embedded smart rock. Secondly, the smart rock embedded with the PZT sensor was designed in order to catch, sensitively, the impedance variations induced by the concrete damage in the anchorage zone. Thirdly, a numerical analysis and experimental tests were performed in order to predetermine the sensitive frequency band for the impedance monitoring. Lastly, an experiment on the anchorage zone embedded with the smart rocks and the surface-mounted PZT sensors was conducted for impedance monitoring under a series of the loading cases, in order to evaluate the feasibility of the smart rock for the detection of internal concrete damage.

From the numerical and experimental analyses, the following remarks can be drawn: (1) the sensitive frequency range of the smart rock is 150–250 kHz; (2) the impedance features showed ignorable changes in the intact state due to the prestress on the smart rock during the concrete’s strength development; (3) the inner concrete damage was successfully detected using the impedance features by the PZT-embedded smart rock.

The experimental results showed the promising applicability of the smart rock for the detection of incipient concrete damage in the prestressed anchorage zone. For example, smart rocks can be embedded during construction, and their impedance responses can be measured and utilized as a baseline. The measured signals are compared, over time, to the baseline signals in order to determine the health of the monitored structure. As a future study, a real-scale experiment on a prestressed anchorage zone should be conducted in order to evaluate the practicality of the proposed smart rock. Furthermore, recent advanced techniques based on machine learning should be employed in order to analyze multiple parameters (e.g., peak frequency, RMSD, and bandwidth) for damage prediction.

## Figures and Tables

**Figure 1 sensors-21-00353-f001:**
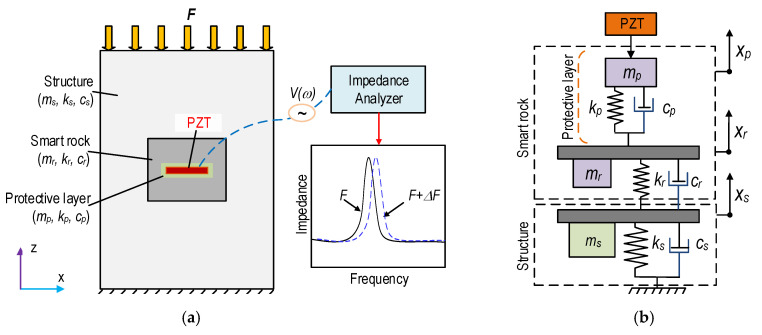
Concept of smart rock-based impedance measurement for concrete structures. (**a**) Concrete structure embedded with smart rock, (**b**) 3-DOF impedance model.

**Figure 2 sensors-21-00353-f002:**
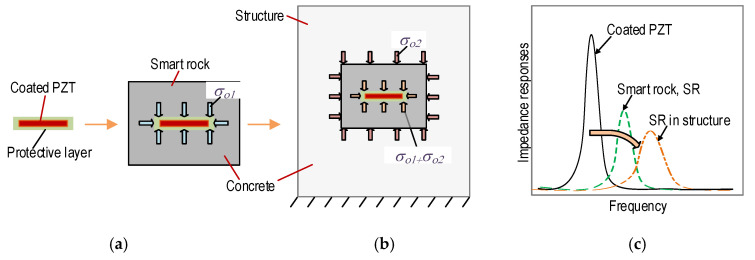
Behaviors of the smart rock under the developing strength of the concrete. (**a**) PZT-embedded smart rock, (**b**) Smart rock embedded in the structure, (**c**) Impedance responses.

**Figure 3 sensors-21-00353-f003:**
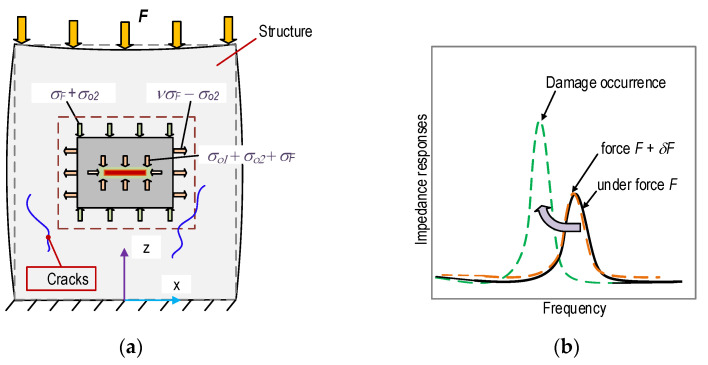
Behaviors of the smart rock under compression. (**a**) PZT-embedded smart rock under compression, (**b**) Variations of the impedance signals.

**Figure 4 sensors-21-00353-f004:**
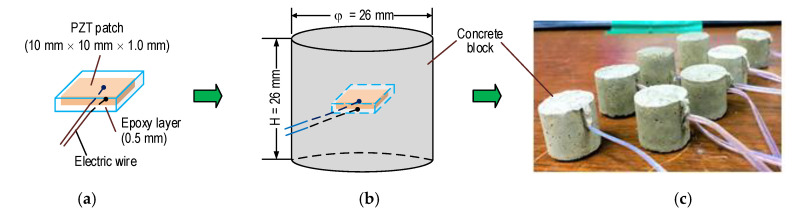
PZT-embedded smart rock for impedance measurement. (**a**) PZT coated with epoxy, (**b**) Smart rock geometry, (**c**) Smart rock samples.

**Figure 5 sensors-21-00353-f005:**
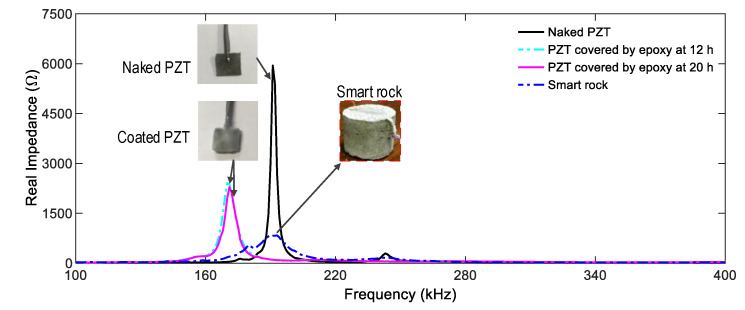
Impedance spectra of naked PZT, coated PZT, and PZT embedded in concrete (smart rock).

**Figure 6 sensors-21-00353-f006:**
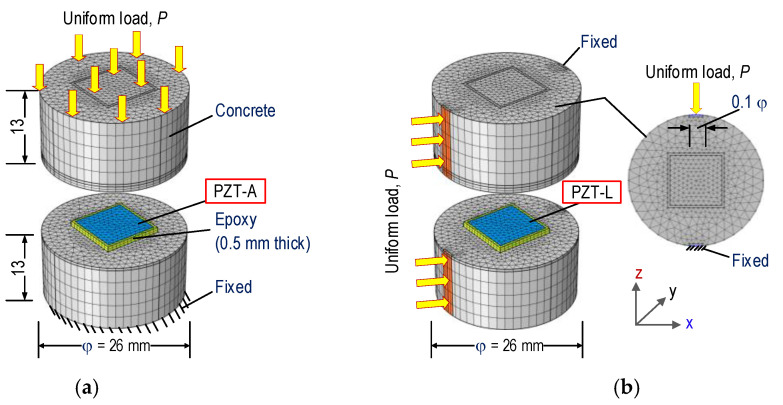
FE models of PZT-embedded smart rock under compression. (**a**) Model 1: effect of axial force, (**b**) Model 2: effect of lateral force.

**Figure 7 sensors-21-00353-f007:**
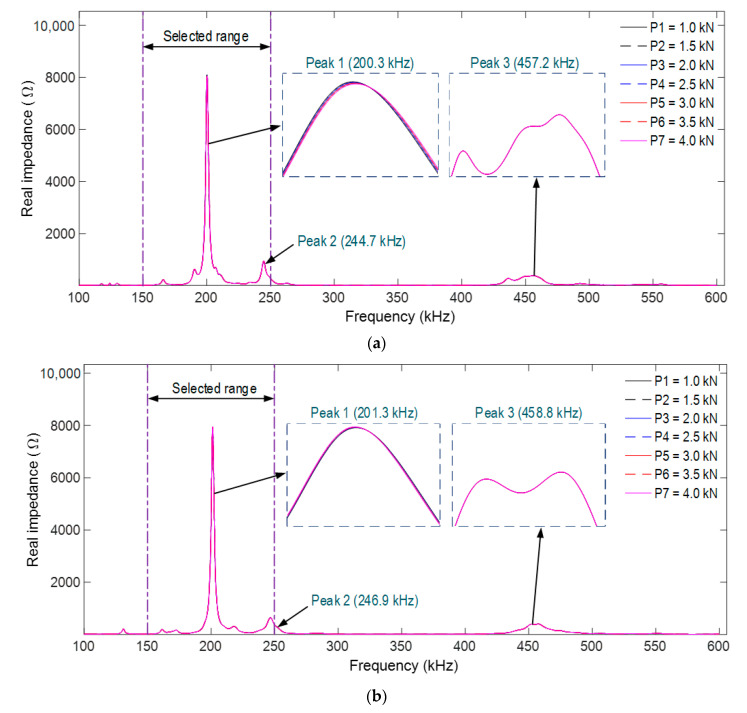
Numerical impedance responses of PZT-embedded smart rock under compression. (**a**) PZT-A under axial compressive force, (**b**) PZT-L under lateral compressive force.

**Figure 8 sensors-21-00353-f008:**
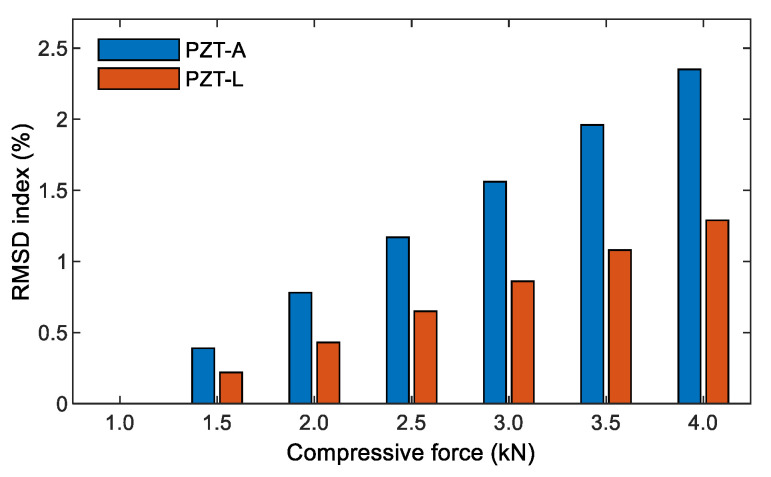
RMSD indices of the numerical impedance responses of the PZT sensors under compression.

**Figure 9 sensors-21-00353-f009:**
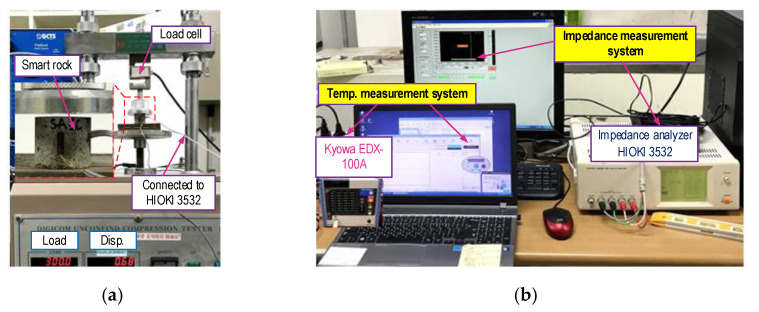
Test set-up and instrumentation for the impedance measurement of smart rock under compression. (**a**) Set-up of the compressive tester, (**b**) Impedance and temperature measurement systems.

**Figure 10 sensors-21-00353-f010:**
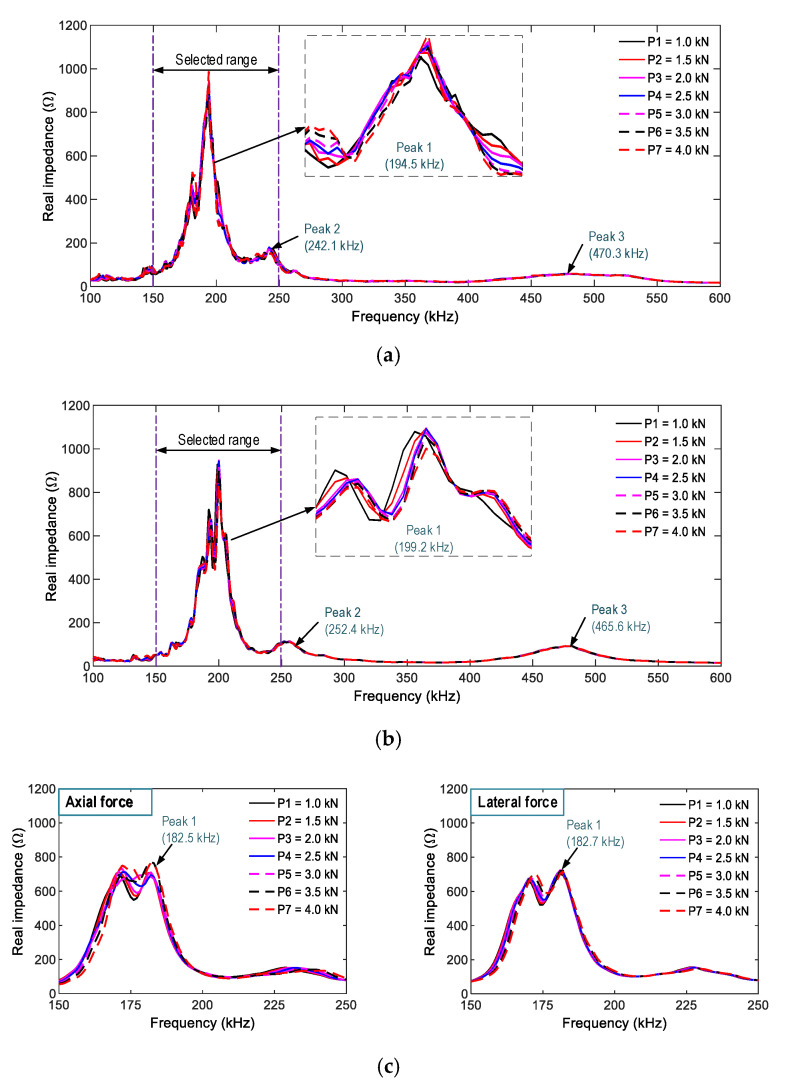
Experimental impedance responses of smart rocks under compression. (**a**) SR-1 under axial compressive force, (**b**) SR-2 under lateral compressive force, (**c**) SR-3 under axial and lateral compressive forces.

**Figure 11 sensors-21-00353-f011:**
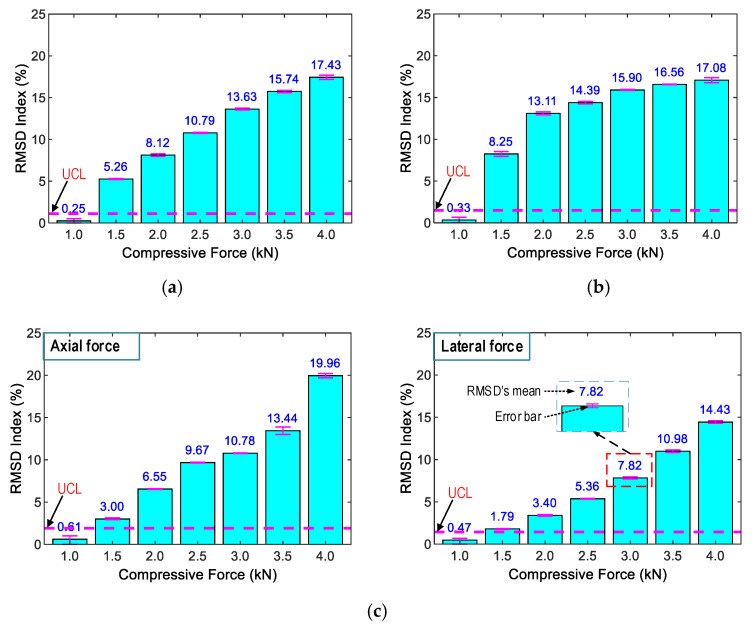
RMSD indices of the experimental impedance responses from the smart rocks under compression. (**a**) SR-1 under axial compressive force, (**b**) SR-2 under lateral compressive force, (**c**) SR-3 under axial and lateral compressive force.

**Figure 12 sensors-21-00353-f012:**
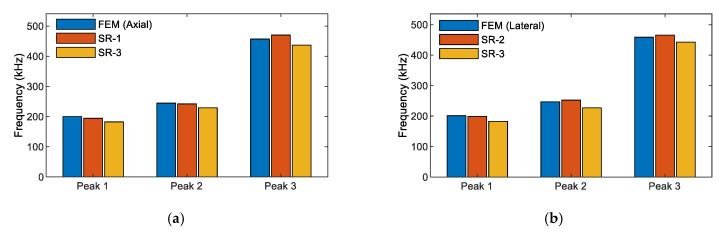
Comparison of the numerical and experimental impedance frequencies of the smart rock. (**a**) Under axial compressive force, (**b**) Under lateral compressive force.

**Figure 13 sensors-21-00353-f013:**
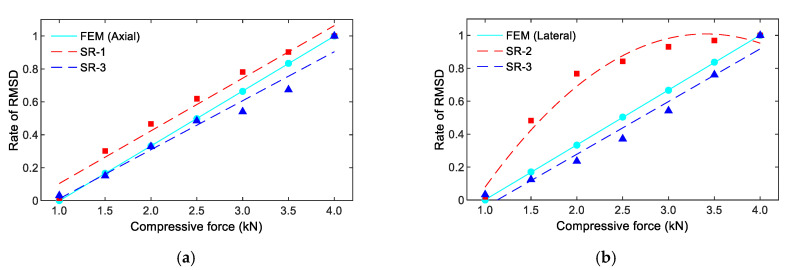
Comparison of the rate of the RMSD indices of the PZT-embedded smart rock. (**a**) Under axial compressive force, (**b**) Under lateral compressive force.

**Figure 14 sensors-21-00353-f014:**
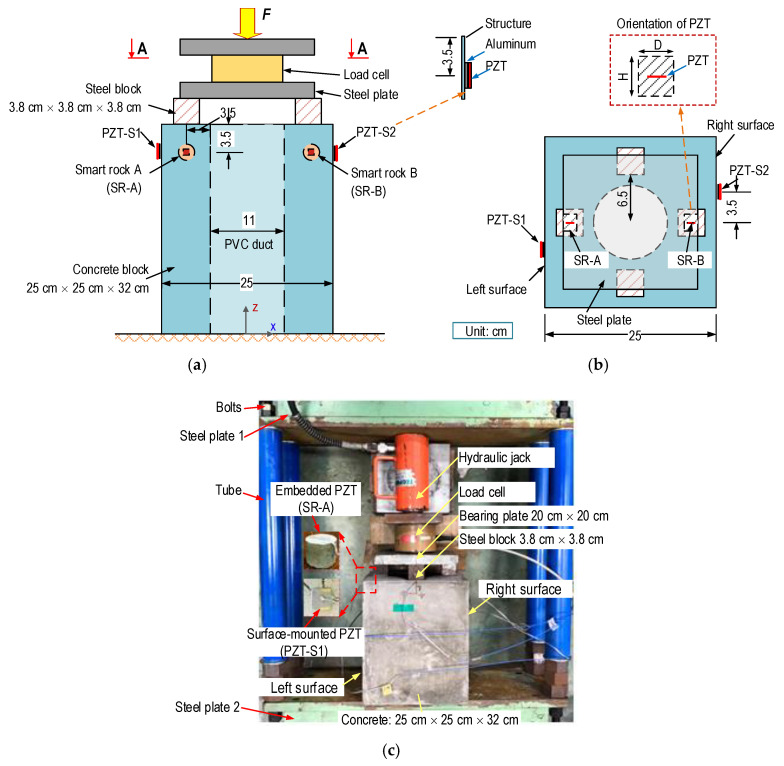
Experimental set-up in the anchorage zone for impedance measurement. (**a**) Side view, (**b**) View A-A, (**c**) Set-up anchorage on the supporting steel frame.

**Figure 15 sensors-21-00353-f015:**
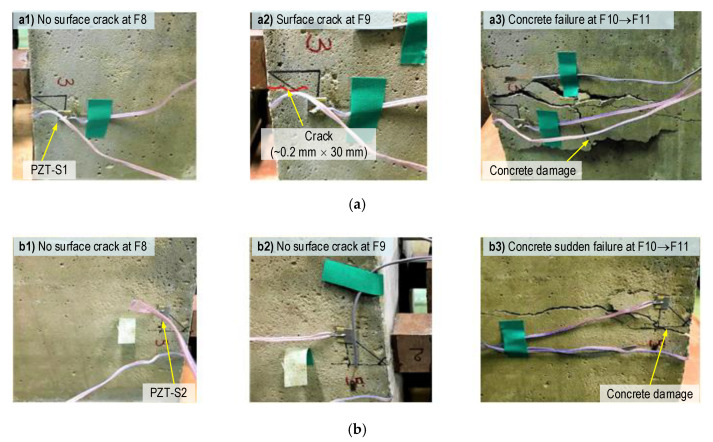
Crack development on the concrete surface of the anchorage zone during the loading process. (**a**) Left side of the anchorage zone (near SR-A), (**b**) Right side of the anchorage zone (near SR-B).

**Figure 16 sensors-21-00353-f016:**
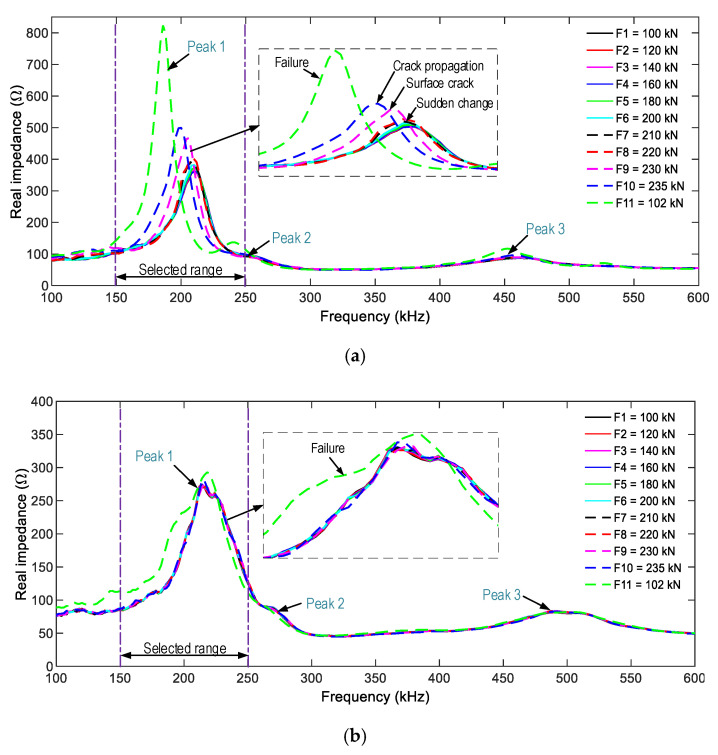
Impedance responses of the PZT-embedded smart rocks in the anchorage zone. (**a**) SR-A, (**b**) SR-B.

**Figure 17 sensors-21-00353-f017:**
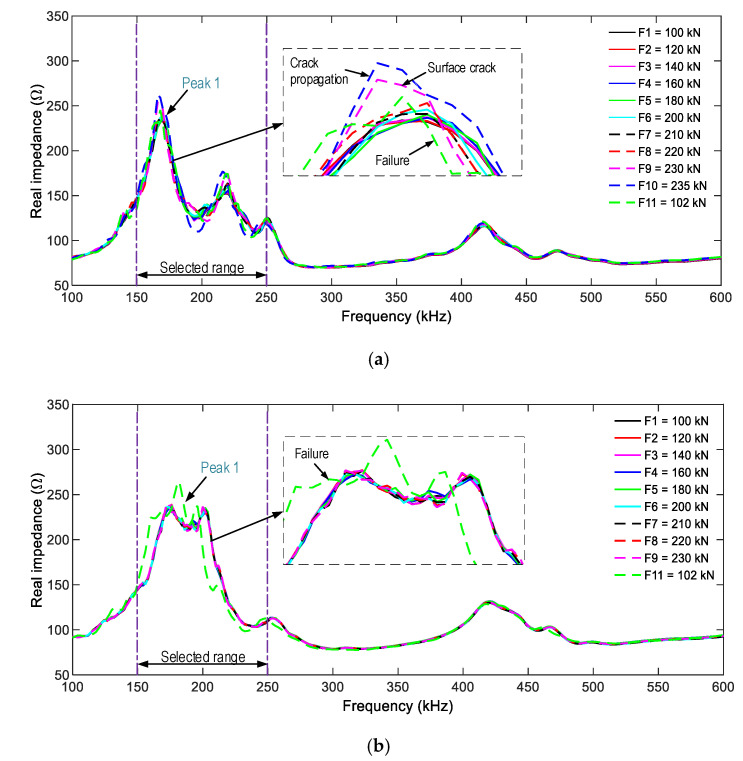
Impedance responses of the surface-bonded PZT sensors on the anchorage zone. (**a**) PZT-S1, (**b**) PZT-S2.

**Figure 18 sensors-21-00353-f018:**
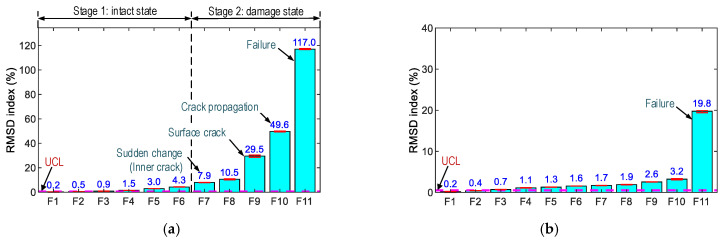
RMSD indices of the impedance responses of PZT-embedded smart rock in the anchorage zone. (**a**) SR-A, (**b**) SR-B.

**Figure 19 sensors-21-00353-f019:**
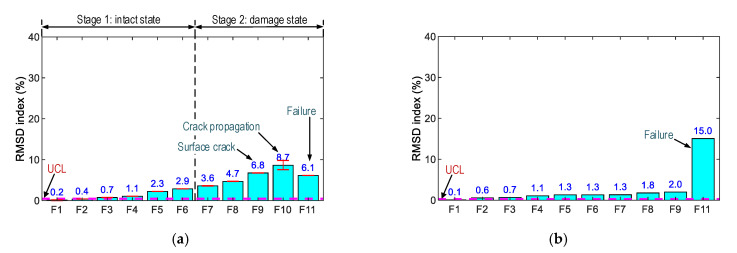
RMSD indices of impedance responses of surface-bonded PZT sensors on the anchorage zone. (**a**) PZT-S1, (**b**) PZT-S2.

**Table 1 sensors-21-00353-t001:** Concrete mixture for the anchorage zone *.

Material	Mass (kg)
Cement	346
Sand	800
Aggregate (D_max_ 25)	997
Water	165

(*) A mixture of cement, sand, and water was used for the smart rock fabrication.

**Table 2 sensors-21-00353-t002:** Material properties of the PZT 5A.

Mass Density, *ρ* (kg/m^3^)	Dielectric Constant $\varepsilon_{}^{T_{33}}(F/m)$	Coupling Constant *d*_31_ (m/V)	Young’s Modulus *E* (GPa)	Damping Loss Factor *η*	Dielectric Loss Factor *δ*
7750	1.53 × 10^−8^	−1.71 × 10^−10^	62.1	0.0125	0.015

**Table 3 sensors-21-00353-t003:** Material properties of the concrete and the epoxy layer.

Properties	Concrete	Epoxy
Young’s modulus (GPa)	23.6	0.74
Poisson’s ratio	0.2	0.3
Mass density (kg/m^3^)	2300	1090
Tensile strength (MPa)	2.3	20.1
Compressive strength (MPa)	23.3	32.3

**Table 4 sensors-21-00353-t004:** List of tests of the smart rocks’ performance under compression.

Test	Name	Force Direction		Applied Load
		Axial	Lateral	
1	SR-1	o	-	P1 (1.0 kN) → P7 (4.0 kN) 0.5 kN incrementin a row
2	SR-2	-	o	
3	SR-3	o	-	
4		-	o	

**Table 5 sensors-21-00353-t005:** Simulation cases of the lab-scale anchorage zone for impedance monitoring.

Case	Applied Force		Concrete Surface’s Crack Development
	F (kN)	Bearing Stress *σ_b_* (MPa) **	
F1 → F6	100 → 200	17.3 → 34.6	- Surface near SR-A: no crack (see Figure 15(a1)). Surface near SR-B: no crack (see Figure 15(b1)).
F7	210	36.4	
F8	220	38.1	
F9	230	39.8	- Surface near SR-A: surface crack initiation, crack size 0.2 × 30 mm (see Figure 15(a2)). Surface near SR-B: no crack (see Figure 15(b2)).
F10	235 *	40.7	- Surface near SR-A: cracks propagation, concrete failure occurred (see Figure 15(a3)). Surface near SR-B: large crack occurred, failure after the other side (see Figure 15(b3)).
F11	102	17.7	

(*) Impedance signals of SR-A, SR-B, and PZT-S1 were measured during the crack propagation on the left surface of concrete. (**) The stress ahead of the steel blocks (bearing plate) is so-called ‘bearing stress’.

## Data Availability

Data available on reasonable request from the corresponding author.

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
