# Peer review of "Piezoelectric Sensor-Embedded Smart Rock for Damage Monitoring in a Prestressed Anchorage Zone"

_sensors, 2021, doi:10.3390/s21020353_

Round 1

Reviewer 1 Report

The paper developed a “smart rock” with a PZT sensor embedded and used the impedance of the “smart rock” to monitor the damage development of a concrete structure under compression. Monitoring the impedance of embedded PZT sensors has been widely and thoroughly studied in the last 20 years. This work is still a good example of using this method for solving an engineering problem although limited theoretical merits were illustrated in this work. The reviewer suggests extending the research to a new sensor design or new data processing algorithm, such as using machine learning with multiple parameters (peak frequency, bandwidth, RMSD, etc.) for damage prediction, rather than using the same method on another engineering application. The paper was well organized and written except for some minor grammar errors. The reviewer has the following comments:

  1. Check the grammar of Line 63-64.
  2. Line 78-79: why applying an impedance-based technique for detecting internal damage in the anchorage area? Reasons like PZT is embedded, real-time, cost-effective which allows multiple sensors, should be explained
  3. Line 165: The reviewer suggests citing the original paper or a representative paper of using RMSD as the damage index, such as Giurgiutiu, Victor, and Craig A. Rogers. "Recent advancements in the electromechanical (E/M) impedance method for structural health monitoring and NDE." Smart Structures and Materials 1998: Smart Structures and Integrated Systems. Vol. 3329. International Society for Optics and Photonics, 1998.
  4. The physical relationship between RMSD and impedance change should be explained in the section.
  5. The reviewer assumes that the UCL is used as the threshold for damage or no damage. This should be mentioned when introducing UCL.
  6. For section 3.1, the impedance spectra of naked PZT, coated PZT, and PZT with concrete in one figure could provide information on how the process affects the PZT impedance. Figure 19 should be here.
  7. Is the mix ingredient of the PZT concrete block the same as the mix ingredient of the anchorage member (Table 1)? The mix ingredients in Table 1 could be converted to mass per unit volume.
  8. The concrete compressive strength in Table 3 is a little lower since currently, most concrete has a compressive strength of more than 28Mpa (>4000psi).
  9. A Scatter curve is suggested in Figure 7, same for Figure 10 and Figure 18. Figure 7 should be plotted with RMSD vs. force (stress). The two subfigures in Figure 7 can be merged to give a better comparison. Seven levels of force had been applied. Why do the results only have four levels of force?
  10. How do you apply the lateral force? Since the PZT concrete block is a cylinder, applying the same force in the lateral direction and axial direction will cause different stress. So, these two scenarios are not comparable. What is the test sequence for SR-3, axial compression first, and then lateral compression? If so, will the axial compression cause some permanent damage to the lateral compression test?
  11. In Figure 9, why are the impedance spectra of SR-3 so different with SR-1, SR-2, and the FE simulation in 6? SR-3 has a different peak frequency and the spectrum is broader.
  12. Line 312-319: Why is the baseline UCL established using the signal at P1? For P1, the sample is still under the load of 1KN, which may not be an intact status. Should the free status (no loading) be used as the baseline?
  13. In Figure 12(b), it is not appropriate to use a linear curve to fit SR-2 (red curve).
  14. In Figure 15(a), a clear peak frequency shift is observed, so does the bandwidth. The reviewer suggests adding this information.
  15. Check the grammar of Line 83-84,302,484,522
  16. Line 485-490: is there a reason that the surface PZT sensors are not sensitive to compression damage, especially to early damage?

Author Response

The authors would like to thank the reviewer for spending precious time on reviewing this manuscript. The manuscript has been improved by reflecting the reviewer’s comments. Please see detailed answers to the reviewer's comments in the enclosed file. 

Thank you very much, and Happy New Year.  

Reviewer 2 Report

The manuscript is generally well organized. The technical content is presented in detail. I just have the following comments and suggestions for the authors to improve their manuscript:

  1. I notice that there are two pairs of piezoelectric sensors used in the manuscript, PZT-A and PZT-L, PZT-s1 and PZT-s2. Are all of them PZT-5A material? The parameters of the piezoelectric sensors should be presented in the manuscript, such as dimensions and resonant frequency. 
  2. In figure 16, how to explain the different impedance responses of PZT-s1 and PZT-s2?
  3. What is the next step of this study? How can the findings in this study help the applications?
  4. The language and grammar of the writing should be carefully checked. Many grammar problems can be found in the manuscript. 

Author Response

(The authors gave the same response as above.)

Reviewer 3 Report

This paper presents an interesting application of a piezoelectric sensor-embedded smart rock as a SHM method. This technique is proposed to catch impedance variations induced by internal concrete damage in the prestressed anchorage zone. Numerical simulations and experimental tests were performed to predetermine the sensitive frequency band for impedance response. Eventually, some experimental tests were conducted under different loading scenarios to evaluate the feasibility of the SHM technique herein proposed.

In the reviewer's opinion, the manuscript is interesting, and the experimental methods are well described. Therefore, the article could be accepted for publication after some minor revision described below.

--------------

As a first hint, the manuscript is written in a good English grammar, only some typos have been found within the text.

In the following a list of suggestions and requests of clarification is given.

Major comments:

(1) It is known that some of the problems in the use of piezoelectric sensors is due to rapid changes of capacitance between conductors. In addition, flexing, twisting or transient impacts on coaxial cables could cause electromagnetic noises in the signal (with a spectrum from few Hz to tens THz). This phenomenon is generally called triboelectric effect and can induce false signals generation.

Many Authors tend to ignore this inconvenience, or they declare that no triboelectric effect is observed during their experimentation, without however providing detailed information on how the latter is eliminated.

It would be very helpful to improve the quality of the manuscript if the Authors spent a few words on this subject.

(2) In Figs. 10 and 18 are reported the RMDS indices of impedance signals. Please, add the error bars in the graphs.

(3) At lines 63-64 of page 2, the Authors write: “…To overcome the issue, the PZT-embedded in a small 63 concrete block (so-called smart aggregate) before employing a monitored structure.”

The sentence is not clear. Please, rephrase this sentence.

(4) At line 252 of page 7, the Authors wite: “It is known the resonant impedance frequency ranges meaningful structural information [11, 252 47].”

The sentence is not clear. Please, rephrase this sentence.

Minor comments:

(5) At lines 181-182 the Authors write: “ Since most piezoelectric materials (e.g., PZT patch) are fragile, it is commonly covered in the form of “smart aggregate” [22, 40] before embedded into concrete members to monitor structural performance…”

Please, give the definition of "smart aggregate" for instance at the end of Introduction.

(6) Line 230, page 6. Please, specify the order of the polynomial interpolation of the tetrahedral elements used for the FEA.

7) Finally, in the Introduction, Authors speak of the use of SHM applications for damage quantification and monitoring. As a suggestion for the Authors, in order to extend the references and make the interest of their manuscript a little wider in the field of SHM, they could cite and briefly comment in the Introduction the following paper:

- Lacidogna G, Piana G, Accornero F, Carpinteri A. Multi-technique damage monitoring of concrete beams: Acoustic Emission, Digital Image Correlation, Dynamic Identification. Constr Build Mater 2020; 242: 118114.

This contribution, recently appeared in the literature, deals with the problem of the damage identification by Acoustic Emission (AE), which is performed by ceramic piezoelectric sensors, and the correlation of the fracture energy with that measured by AE.

The paper, moreover, considering plain concrete specimens under three-point bending tests, not only demonstrates the effectiveness of the combination of AEs and Digital Image Correlation (DIC) to identify damage in concrete structural elements, but also of Dynamic Identification (DI) procedures.

The final comment of the referee is that the manuscript is suitable for publication after the above suggestions have been implemented.

Author Response

(The authors gave the same response as above.)

Round 2

Reviewer 1 Report

The reviewer only has one comment on the revised manuscript (see below) and the current version meets the criteria of publication.

The references do not have a consistent style. For example, ref 8, 14, 23 capitalize each word in the title, but others are sentence case.